# Stochastic ordering of complexoform protein assembly by genetic circuits

**Mikkel Herholdt Jensen**[1]*, **Eliza J. Morris**[1], **Hai Tran**[2], **Michael A. Nash**[3,4], **Cheemeng Tan**[5]*

**1** Department of Physics and Astronomy, California State University, Sacramento, California, United States of America, **2** Department of Chemistry, California State University, Sacramento, California, United States of America, **3** Department of Chemistry, University of Basel, Basel, Switzerland, **4** Department of Biosystems Science and Engineering, ETH Zurich, Basel, Switzerland, **5** Department of Biomedical Engineering, University of California, Davis, California, United States of America

* mikkel.jensen@csus.edu (MHJ); cmtan@ucdavis.edu (CT)

**Data Availability Statement:** All data are included in S1 Data.

**Funding:** The work is supported by Human Frontier Science Program (FR) (RGY0080/2015), California State University, Sacramento, Department of

## Abstract

Top-down proteomics has enabled the elucidation of heterogeneous protein complexes with different cofactors, post-translational modifications, and protein membership. This heterogeneity is believed to play a previously unknown role in cellular processes. The different molecular forms of a protein complex have come to be called "complex isoform" or "complexoform". Despite the elucidation of the complexoform, it remains unclear how and whether cellular circuits control the distribution of a complexoform. To help address this issue, we first simulate a generic three-protein complexoform to reveal the control of its distribution by the timing of gene transcription, mRNA translation, and protein transport. Overall, we ran 265 computational experiments: each averaged over 1,000 stochastic simulations. Based on the experiments, we show that genes arranged in a single operon, a cascade, or as two operons all give rise to the different protein composition of complexoform because of timing differences in protein-synthesis order. We also show that changes in the kinetics of expression, protein transport, or protein binding dramatically alter the distribution of the complexoform. Furthermore, both stochastic and transient kinetics control the assembly of the complexoform when the expression and assembly occur concurrently. We test our model against the biological cellulosome system. With biologically relevant rates, we find that the genetic circuitry controls the average final complexoform assembly and the variation in the assembly structure. Our results highlight the importance of both the genetic circuit architecture and kinetics in determining the distribution of a complexoform. Our work has a broad impact on our understanding of non-equilibrium processes in both living and synthetic biological systems.

## Author summary

Multiple protein subunits can come together to form protein complexes that play critical functional roles in a cell. Recent advancement in measurement technologies has revealed tremendous variation in the members of protein complexes. The recent results motivate

Physics and Astronomy Chien Hu Research Award Program, the Edwin L. Iloff Endowment, and NSF (1808237). The funders had no role in study design, data collection and analysis, decision to publish, or preparation of the manuscript.

**Competing interests:** The authors have declared that no competing interests exist.

further research into the importance and the underlying mechanisms of the variation. Here, we study the arrangement of genes as a key factor that modulates the variation of protein complexes. We run computer simulations to investigate how various reaction parameters control the variation of a protein complex. Finally, we extend our framework to study the variation of an enzymatic complex that digests cellulose. Our work has a broad impact on the understanding of protein-complex assembly and set up the new research direction about the variation of protein complexes.

## Introduction

Proteins are synthesized in specific orders to assemble large protein complexes, such as micro-tubule, proteasome, ribosomes, and cellulosome. These protein complexes are assembled both inside and outside cells through the coordination of gene expression, protein transport, and binding processes. Prior work has been assuming that protein complexes have a homogeneous composition of protein members. Yet, recent top-down proteomics shows that protein complexes can compose of different cofactors, post-translational modifications, and protein membership [1–3]. The different molecular forms of a protein complex have come to be called the complex isoforms or complexoforms [1]. For example, recent work shows that the yeast homo-tetrameric FBP1 complex can co-exist with 0 to 4 phosphorylated amino acids [4]. Bacteria cel-lulosomes are also found to exist in heterogeneous compositions [5–11]. Furthermore, a recent computational study [12] has investigated the formation of protein complexes using existing data on protein-protein interaction networks. This prior work shows that the composition of a protein complex can drift over time even when the simulation starts from the same initial condition. The work suggests that other cellular mechanisms must exist to prevent the compositional drift of some protein complexes.

While the elucidation of complexoforms is now possible, finding the mechanisms that control the distribution of complexoforms is challenging. There are two competing paradigms for understanding the formation of complexoforms. A classical but unproven paradigm assumes that temporal ordering of protein synthesis is essential for accurate assembly of protein complexes. Yet, most biochemical studies mix proteins in a test tube to study protein complex assembly, neglecting the underlying cellular networks that control synthesis and transport of the proteins. This approach suggests a second, contradicting paradigm: the underlying cellular networks are not essential for the assembly of quaternary protein complexes.

One way to resolve the contradiction is to examine genome sequences for the spatial arrangement of genes in the genome: if genes are arranged in specific patterns that correlate with their synthesis order or the assembly sequence of protein complexes, then the underlying gene networks may be essential for the protein assembly. Indeed, a study of the human genome shows that housekeeping genes expressed in most tissues show strong clustering [13]. The data suggest that the ordering of protein synthesis may be conserved for critical protein complexes [14]. For non-critical protein complexes, either each cell-type evolves its protein synthesis order to achieve unique complexoforms, or the protein synthesis order is not an important factor.

A central question currently challenging the field is, therefore, what underlying rules govern protein-synthesis or assembly order, and the limit of the rules in controlling the distribution of a complexoform. Here, we use stochastic computational simulations to model protein expression and assembly from the bottom-up. We first simulate a three-protein model system to investigate how complexoform composition is affected by the timing of gene transcription,

mRNA translation, and protein transport, as well as the gene circuit architecture. Due to the small number of chemical agents and proteins often involved in protein assembly processes, ordinary equilibrium statistical mechanics approaches are not expected to adequately describe the eventual protein assembly outcome. We show that genetic circuits, physical transport, and binding kinetic rates all modulate the distribution of complexoform. Comparison of the stochastic simulations to deterministic solutions reveal two distinct kinetic regimes: a slow equilibrium regime in which the average structure is well predicted by equilibrium statistical mechanics, and a fast nonequilibrium regime in which small-number statistics leads to structures far from equilibrium predictions. This work points to new and previously unappreciated mechanisms of modulating the distribution of a complexoform. Our work also highlights the importance of both gene circuit architecture and nonequilibrium stochasticity in regulating the formation of complexoforms. Our work provides a possible resolution of the gene-circuit vs. protein-only paradigms: gene-circuit-based regulation of complexoform distribution may be important only under the select kinetic regime and cellular context.

## Results

### The type of genetic circuit modulates complexoform assembly

Protein expression and assembly typically involve a relatively small number of each type of molecule (<1000 per type of protein [15]). In such cases, equilibrium statistical mechanics may not accurately describe the process of protein expression and assembly, since the ensemble of reactants is insufficiently small. Thus, the process becomes inherently stochastic in nature. Here, we use a stochastic model to examine the heterogeneity in protein expression and assembly.

Our model system consists of a genetic circuit expressing two types of proteins, denoted *X* and *Y*. The proteins are exported to a sub-cellular location and then bind to scaffold proteins, each consisting of two docking sites. This model system allows us to investigate the role of timing and genetic circuit architecture in modulating the protein assembly, irrespective of the exact number of types of proteins and binding sites. In this model, the binding between the proteins creates a protein complex (Fig 1). The proteins compete for 10 scaffold proteins. The

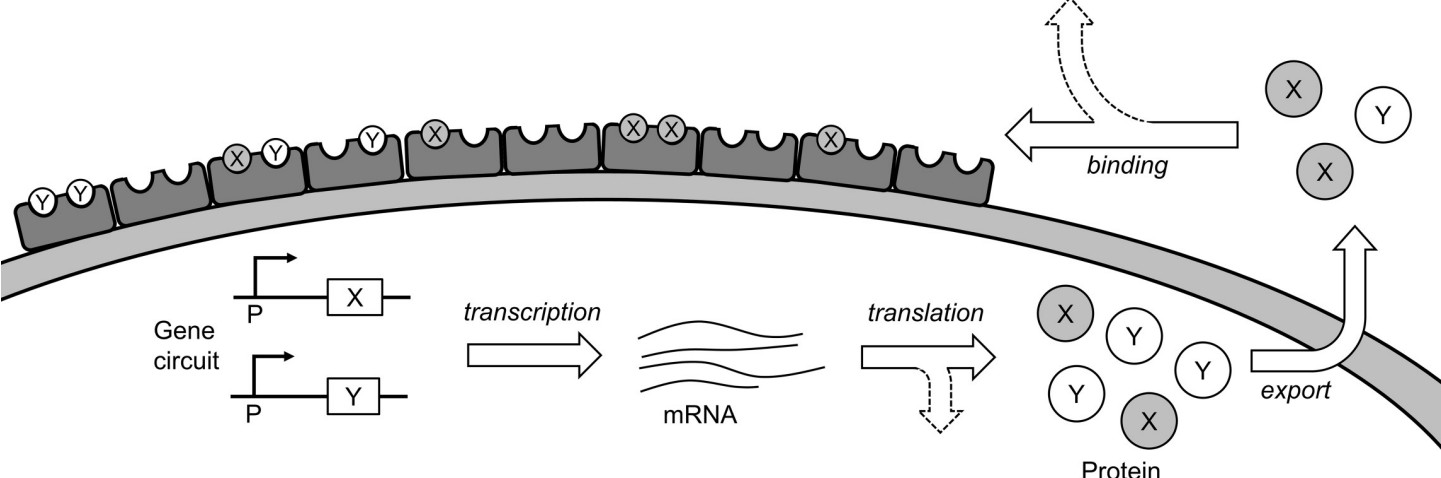

**Fig 1. Illustration of the complexoform assembly model system.** Simulated processes and rates are indicated by arrows and italicized text. In the model, genetic circuitry coding for either protein X or protein Y is initiated by a promoter and transcribed to mRNA. Translational machinery then produces proteins X and Y, which are exported and compete for a limited number of scaffold proteins, each with two docking sites. The simulation also incorporates a rate of loss of mRNA and external protein. Generic simulations are carried out with 10 external scaffold proteins (i.e., 20 docking sites).

final complexoforms can thus consist of *XX*, *XY*, or *YY*. When a small number of molecules are involved in each process, the exact ratio of each is stochastically determined. To establish the average distribution of complexoform, simulations are run 1,000 times for each genetic circuit and each set of parameters. This number of simulations is found to be sufficient to reach a reproducible average to within an uncertainty of 1 percentage point of *XX*, *XY*, and *YY*.

We set up four common types of genetic circuits expressing two proteins, *X* and *Y*. The series uncoupled circuit expresses the two proteins from the same operon, but the translation occurs at two separate ribosome binding sites. The series coupled circuit expresses the two proteins from the same operon and the translation occurs through a single ribosome binding site. The cascade circuit expresses the two proteins from two separate promoters linked in a cascade. Finally, the parallel circuit expresses the two proteins independently from two identical promoters.

We first investigate the role of the specific genetic circuit architecture in determining the protein assembly by assuming that the rate constants are identical. This assumption helps to isolate the effect of gene circuit architecture on the formation of complexoform. It will be relaxed in subsequent simulations. Each genetic circuit modulates the distribution of the complexoforms differently (Fig 2). For the cascade, series uncoupled, and series coupled circuits, the final complexoforms are predominantly *XX*. This trend is most evident for the series coupled circuit (84% *XX*, 15% *XY*, 1% *YY*) and the cascade circuit (86% *XX*, 10% *XY*, 4% *YY*). The results arise because these circuits produce the mRNA for *X* or protein *X* first. For the series uncoupled circuit, protein *X* is less prevalent in the final complexoform, but still exceeds protein *Y* (45% *XX*, 38% *XY*, 17% *YY*). The parallel circuit stochastically produces similar amounts of *X* and *Y* in the final complexoform (45% *XX*, 12% *XY*, 43% *YY*). These results indicate that even with identical rate constants and initial concentrations of all the reactants, the genetic circuit architecture can significantly modulate the final distribution of the complexoform.

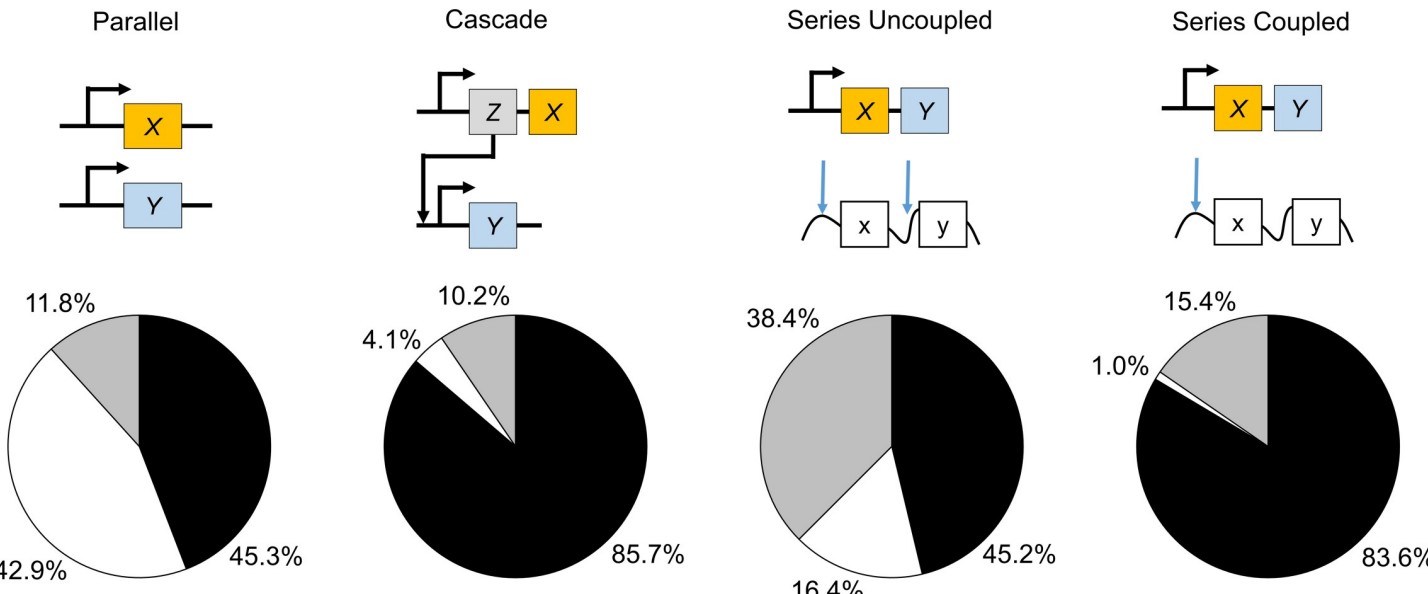

**Fig 2. Genetic circuit architecture modulates complexoform assembly.** Four different genetic circuits expressing proteins X and Y (parallel, cascade, series uncoupled, and series coupled) are simulated with identical first-order rates of transcription (1 t$^{-1}$), translation (0.1 [C]-1t$^{-1}$), export (1 t$^{-1}$), and binding (1 [C]$^{-1}$t$^{-1}$) to 10 scaffold proteins each with 2 docking sites. The pie charts indicate the average outcomes from 1,000 stochastic simulations, showing the percentage of XX (black), YY (white), and XY/YX complexoforms (grey). The simulation results demonstrate that the genetic circuit architecture controls the final protein assembly structure.

### Reaction rates of both protein expression and physical processes modulate complexoform assembly

Next, we determine the impact of the reaction constants on protein assembly. We simulate each genetic circuit while individually varying the translation rate, the export rate, and the binding rate. For comparison, we also solve each condition deterministically. Fig 3 shows the mean distribution of the complexoform from stochastic simulation (circles) and deterministic solutions (lines). In all cases, the kinetic rates modulate the final distribution of the complexoform. Slow translation, export, or binding all result in a distribution of complexoform of

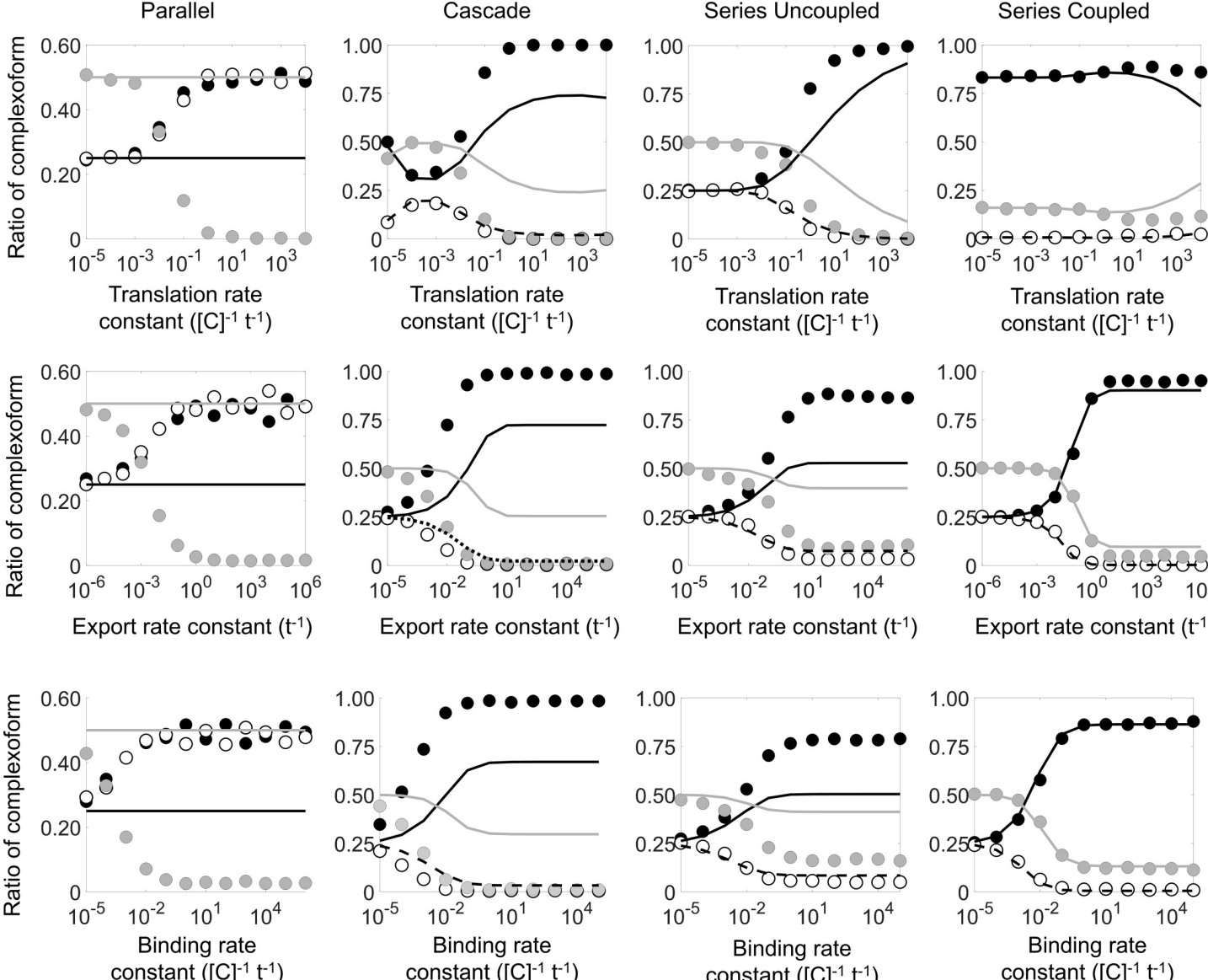

**Fig 3. Computational simulations of complexoform assembly using a generic model system.** Stochastic solutions (XX: black circles; YY: white circles; XY: grey circles) and deterministic solutions (XX: solid black line; YY: dashed line; XY: solid grey line) for parallel (first column), cascade (second column), series uncoupled (third column), and series coupled (fourth column). For the parallel circuit, the deterministic solutions for XX and YY overlap perfectly, so only one line is visible on the graph. For each circuit, variations in translation rate, protein export rate, and binding rate reveal an "equilibrium" regime, in which the stochastic and deterministic solutions are in relatively good agreement, and a "non-equilibrium" regime, in which the stochastic and deterministic solutions disagree. More stringent genetic circuits (e.g., the series coupled circuit) exhibits a lower discrepancy between the stochastic outcome and the deterministic solution.

roughly 25% *XX*, 50% *XY*, and 25% *YY*. Our results show that varying the kinetic rates in the stochastic model alters the distribution of the complexoform, demonstrating that the genetic and physical rate constants, as well as the exact genetic circuit architecture, can modulate the eventual protein assembly. We note that the rate of mRNA degradation also alters the distribution of the complexoform, but that the effect is captured by altering the rate of translation (i.e., varying the ratio of the translation rate to the mRNA degradation rate), as noted in S3 Appendix. In contrast, the promoter binding rate, while changing the deterministic solutions, does not affect the stochastically predicted distribution of the complexoform (see S4 Appendix).

As any one of the rates is increased, the stochastic distribution of the complexoform begins to deviate from the equilibrium outcome of equal amounts of *X* and *Y* in the assemblies. For the parallel circuit, any single stochastic simulation produces either predominantly *XX* or *YY* assemblies, each with a 50/50 chance, and has a very low chance of producing any heterogeneous *XY* complexoforms. Since the structure of the cascade, series uncoupled, and series coupled genetic circuits all favor protein *X* expression before protein *Y*, higher kinetic rates tend to produce *XX* complexoform in these circuits. In contrast, if a sufficiently large number of both types of proteins reach the scaffold protein before the docking sites are occupied, any binding site has an equal chance of binding either protein *X* or *Y*, and the four different complexoforms (*X+X*, *Y+Y*, *X+Y*, and *Y+X*) have an equal chance of occurring. A homogeneous *XX* or *YY* should, therefore, each occur 25% of the time, while a heterogeneous *XY* or *YX* assembly should account for 50% of the final scaffold proteins. This 25/50/25 distribution in the final protein complex is seen in almost all cases when the rates are sufficiently slow (Fig 3). For example, a sufficiently slow rate of protein export results in the combinatorially predicted the 25/50/25 distribution for all four types of genetic circuits (Fig 3, second row). In this case, the protein complex is independent of the underlying genetic circuit architecture: even though the cascade, series uncoupled, and series coupled circuits all synthesize protein *X* before protein *Y*, a bottleneck in a subsequent physical process ensures that the amounts of *X* and *Y* have time to reach similar levels before accessing the docking sites.

## Kinetic rates drive either equilibrium or non-equilibrium assembly

Our simulation results suggest that the protein assembly process can be divided into two fundamental limiting time scales: One slow regime, in which the slow kinetic rates allow for equal amounts of protein *X* and *Y* to access the docking sites, and a fast regime, in which one protein undergoes rapid expression, transport, and binding, thus dominating the complexoform. If the assembly process is sufficiently slowed down by one or more rate-limiting steps, the protein assembly process can be treated as a problem of equilibrium statistical mechanics; however, this is not the case if the protein expression and assembly process are sufficiently rapid.

To better quantify these two limiting time scales, which we term the "equilibrium" and "non-equilibrium" regimes, we numerically solve the set of coupled differential equations deterministically and compare our solutions to the stochastic simulation results. In the deterministic approach, the assembly is not limited by a small number of reactants in the system, and the deterministic numerical solutions, therefore, represent the complexoform in the limit of large-number statistical mechanics. If the stochastically driven protein assembly reaches an equilibrium state, the stochastic simulations and numerical solutions should agree. In contrast, a discrepancy between the stochastic simulations and deterministic numerical solutions would suggest that the protein assembly is not well described by predictions from large ensemble statistical mechanics, but instead is governed by non-equilibrium, small-number statistics.

The predictions from the deterministic solutions are shown together with the stochastic results (Fig 3). The discrepancy between the stochastic simulation average and the

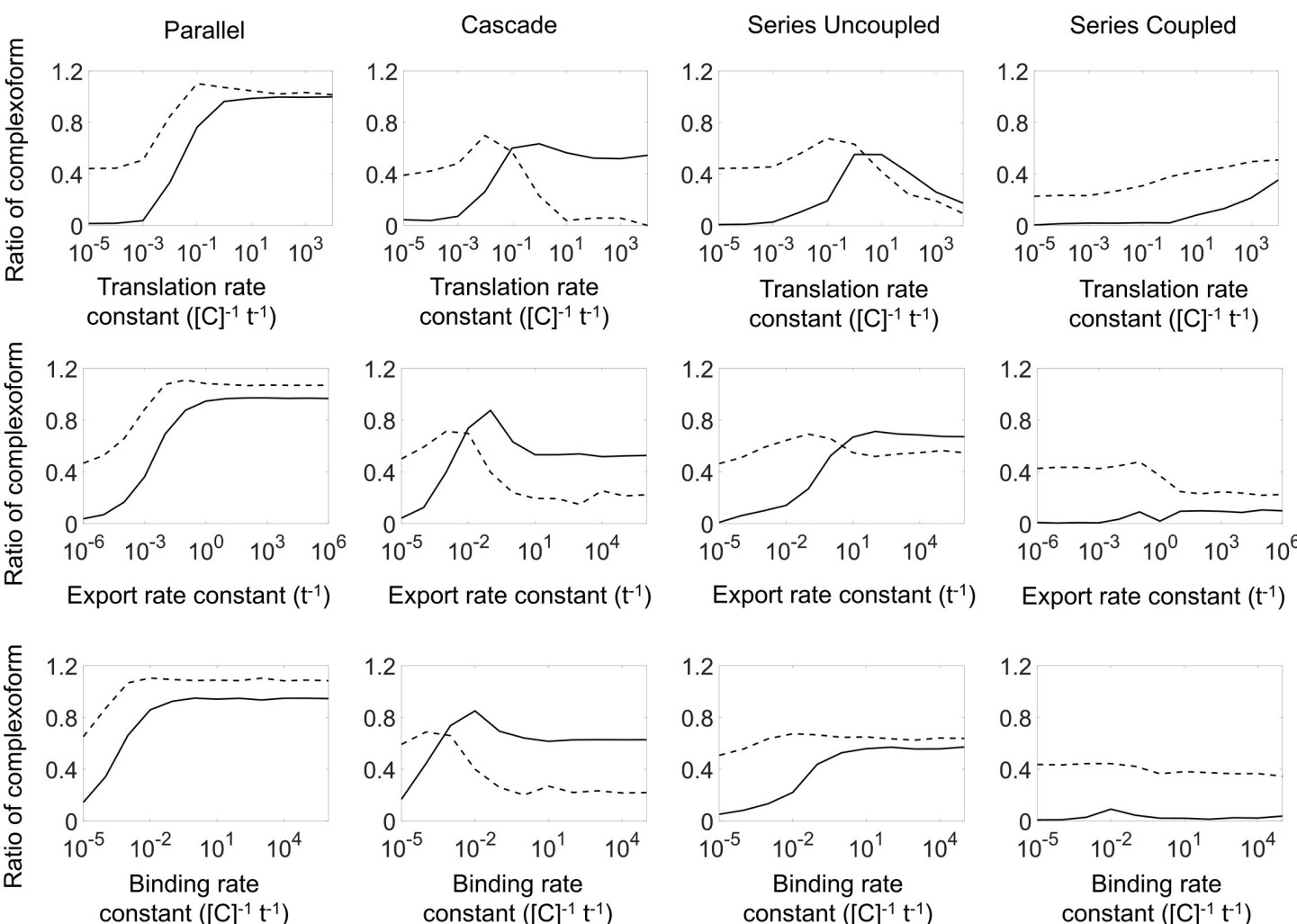

**Fig 4. The discrepancy between the deterministic and stochastic solutions, and stochastic variation in protein assembly structure for four stochastically simulated genetic circuits.** The discrepancy between the stochastic simulations and deterministic solutions (solid line) tends to be greater when rates are high and assembly proceeds quickly, suggesting that the protein assembly in this limit is not accurately represented by the corresponding deterministic solution. However, when assembly is slow, the predicted assembly matches the equilibrium prediction from mixing large amounts of the constituents X and Y, and all circuits yield similar protein assembly products. Circuits with a more controlled sequence of expression (e.g., the series coupled circuit) exhibit an overall better agreement with the deterministic solution. The stochastic variation in assembly (dashed line) also varies between the different circuits, suggesting that the type of genetic circuit, as well as the kinetics of assembly, both play into determining the variability of the assembled structure.

deterministic solution is calculated as the sum of the differences for each of the three types of complexoform assemblies (*XX*, *YY*, or *XY/YX*) (Fig 4) together with the variation in the assembly structure from the stochastic simulations. The stochastic simulations agree with the deterministic solutions in all cases and for all genetic circuits when any one of the kinetic rates is sufficiently slow, corresponding to the equilibrium regime. The equilibrium regime does not imply that all complexoform assemblies would be identical. Indeed, in the equilibrium regime, the complexoform is still stochastically determined and varies from simulation to simulation, as is evident from the non-zero stochastic variation at slow rates (Fig 4, dashed lines). However, in the equilibrium regime, the *average* stochastic outcome is well predicted by treating the problem as a large-ensemble deterministic system. In contrast, when any one of the rates is increased, the stochastic average deviates from the deterministic solution, and the deterministic equilibrium solution no longer accurately predicts the average complexoform from the

stochastic simulation, corresponding to the non-equilibrium regime. The transition from the equilibrium to the non-equilibrium regime is seen as a rise in the discrepancy between the stochastic simulation average and the deterministic solution (Fig 4, solid lines). In the non-equilibrium regime, the protein assembly arrests into a non-equilibrium structure as vacant docking sites are depleted before reaching the equilibrium structure predicted by the deterministic solution. The system transitions between the equilibrium and non-equilibrium regimes when either the genetic expression rates or the physical rates of protein export and binding are altered, and generally coincides with a peak in the stochastic variation in the protein assembly structure. A notable exception is observed for the simulations of the series coupled genetic circuit. Since this circuit imposes the most stringent structure on both the transcription and translation of the proteins, the average stochastic structure agrees well with the deterministic solution, even as the protein export or binding rate is increased.

Our results also show that altering any of the rates of the system, in turn, changes the average total time to fully occupy the available docking sites, which we term time to assembly ($T_{asb}$). The average fraction of *XX*, *XY*, and *YY* complexoform in the stochastic simulation is shown as a function of the average $T_{asb}$ in Fig 5. Contrary to our expectation, the fraction of each complexoform falls onto a single curve, regardless of which of the rates is being changed. Although creating an earlier bottleneck by slowing down the earlier processes, such as translation, results in the longest $T_{asb}$, the total $T_{asb}$ is highly correlated with the complexoform composition, and the average assembly outcome from the stochastic simulation can be correlated to a single parameter, the $T_{asb}$. Here again, however, the series coupled genetic circuit is an exception; when the translation rate in this genetic circuit is lowered, the sequential architecture of the circuit results predominantly in *XX* complexoform. This result again suggests that the gene circuit architecture can exert a high level of control over the protein assembly structure, even when the assembly is slow, and a large number of proteins are involved in stochastically forming the structure.

## Comparisons to other biological model systems

To compare our modeling results to typical rates and concentrations in biological systems, and to determine whether protein assembly in biological systems generally fall in the equilibrium or non-equilibrium regime, we run our computational models with typical biochemical and physical rates of protein expression, export, and binding for bacterial cellulosome. This system is modeled as a scaffold protein with 10 binding sites to which proteins *X* and *Y* can bind. Rates representative of the bacterial cellulosome complexoform, as well as relevant references, are summarized in Table 1. Since it is difficult to estimate the exact external per-molecule binding rate and loss rate of individual proteins, we simulate cellulosome assembly with a range of possible protein binding rates while keeping the loss rate fixed, thus varying the ratio of binding to loss rate for the external protein. Each condition is simulated 1,000 times to generate a histogram of the outcomes, as quantified by the number of binding sites on the scaffold protein occupied by protein *X* (the rest being occupied by protein *Y*; see Fig 6). From these simulations, we quantify both the variation in stochastic assembly, as well as the discrepancy between stochastic simulations and deterministic solutions. The simulation results for each of the four genetic circuits (parallel, cascade, series uncoupled, and series coupled) are summarized in Fig 7, with each of the translation rate, protein export rate, and protein binding rate being varied about the estimated physiological value from literature and previous experimentally reported values.

Our simulations show that, as observed for the generic model system, the type of genetic circuit exerts a great deal of control over the eventual protein assembly. For example, a parallel

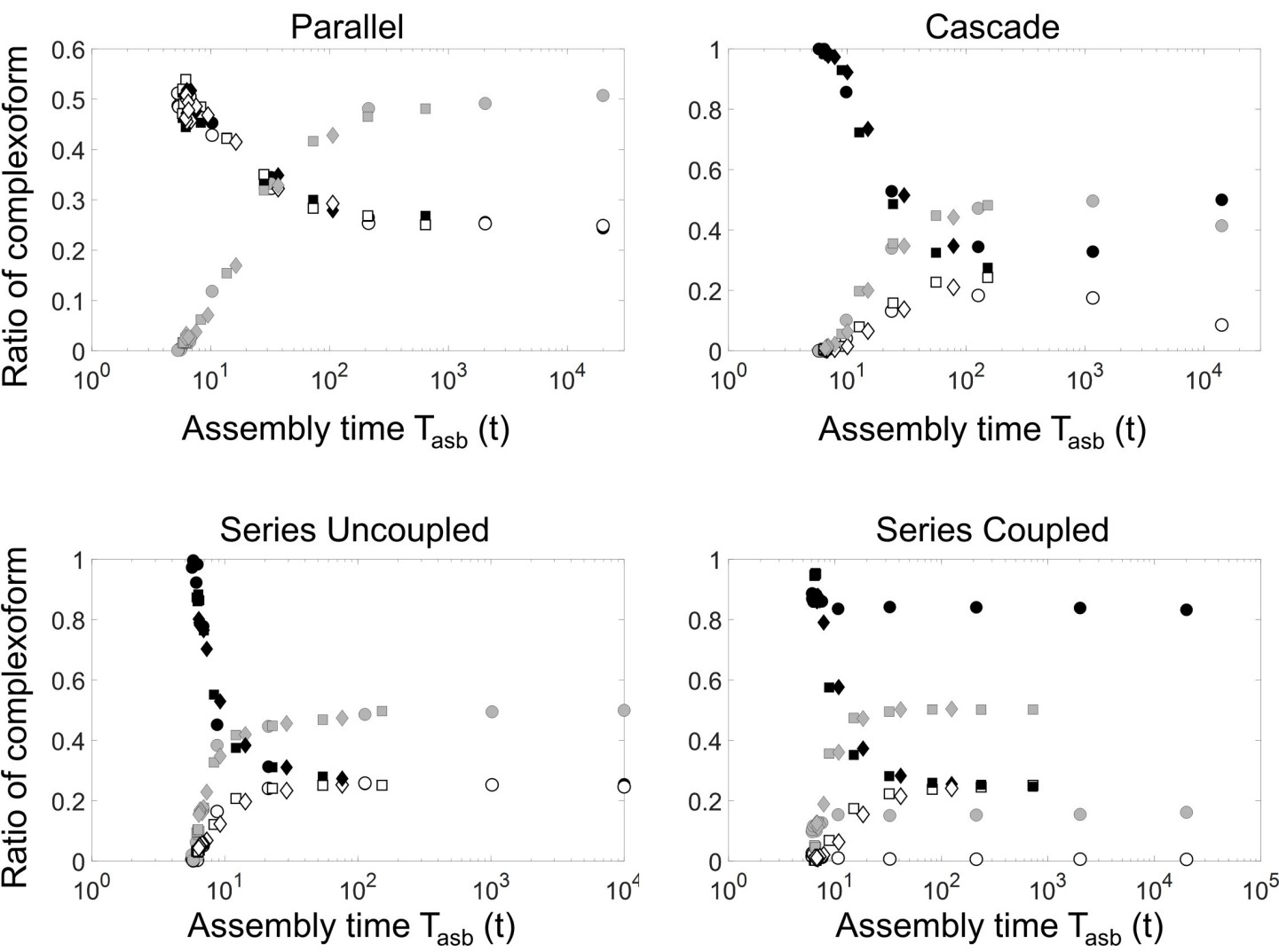

**Fig 5. The genetic circuits generate similar assembly times, regardless of which rate is varied.** Shown is the stochastically simulated average fraction of XX (black), YY (white), and XY complexoform (grey) in the protein assemblies when varying translation rate (circles), protein export rate (squares), and protein binding rates (diamonds). Each circuit exhibits a transition from the fast non-equilibrium regime to the slow equilibrium regime, although the exact transition time is dependent on the type of genetic circuit. A notable exception is the series coupled circuit, indicating that the highly sequential architecture of this circuit modulates the protein assembly structure in both the fast and slow regimes.

circuit will result in an average assembly of 50% *X* and 50% *Y* (Fig 7, column 1), whereas the cascade circuit and the series circuits are weighted towards *X* to varying degrees (Fig 7, columns 2, 3, and 4). As was also observed for the generic model, the parallel genetic circuit exhibits the highest degree of variation in the assembly outcome, whereas the more structured genetic circuits have lower variability in the final assembly composition. One major difference between the generic model (with arbitrary rates) and the cellulosome model is that whereas the stochastic simulations of the generic model would often deviate significantly from the deterministic solutions, the simulations with the rates for the cellulosome system exhibits virtually no such deviations between the stochastic average and the deterministic solution. One exception to this is the parallel circuit, where a very fast rate of expression could have resulted in an assembly consisting of either *X* or *Y* (see also Fig 6C). Overall, however, the model using biological cellulosome rates predicts good prediction between the deterministic and stochastic

**Table 1. Rates and concentrations used in simulating cellulosome assembly.**

| | Value | References |
|---|---|---|
| **Rates** | | |
| $k_{pro}$ (RNAP promoter binding) | $5.6 \cdot 10^7$ M$^{-1}$s$^{-1}$ = 0.09 s$^{-1}$ | [16] |
| $k_{pro-}$ (RNAP promoter unbinding) | 0.20 s$^{-1}$ | |
| $k_{gene}$ (RNAP synthesis rate constant) | 0.36 s$^{-1}$ | [17] |
| $k_m$ (transcription) | 0.03 s$^{-1}$ | [17–20] |
| $k_{mrna-loss}$ (mRNA degradation) | 0.002 s$^{-1}$ | [21] |
| $k_p$ (translation) | 0.03 s$^{-1}$ | [19,20,22] |
| $k_{out}$ (diffusive protein export) | 0.5 s$^{-1}$ | [23–24] |
| $k_{bind}$ (protein association rate) | $1 \cdot 10^6$ M$^{-1}$s$^{-1}$ = 0.002 s$^{-1}$ | [25–26] |
| $k_{bind-}$ (protein dissociation rate) | 0 | |
| $k_{out-loss}$ (protein loss) | 0.0002 s$^{-1}$ | [27] |
| **Reactants** | | |
| P (promoter DNA) | 1 | |
| RNAP (RNA polymerase) | 100 | [28] |
| Rib (ribosome) | 5 | [29–30] |
| Docking sites per scaffold protein | 10 | [5] |

Note: Two proteins are expressed and bind to an external scaffold with 10 binding sites. Intracellular molar concentrations are converted to molecules per cell assuming that 1 M is equal to $6 \cdot 10^8$ molecules per cell. Note that the protein assembly process is assumed to be irreversible for the purposes of this simulation, whereas actual measurements in the literature does show that the complex does disassemble over time.

simulations, although the stochastic simulations still predict significant variability in the final protein assembly. Interestingly, the simulations also reveal that the cellulosome kinetic rates place the system at the transition between the fast and slow limits seen in the generic model. For example, in the cascade genetic circuit, the biological rates predict an average assembly

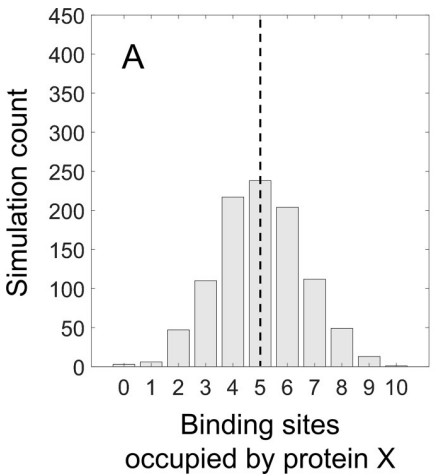 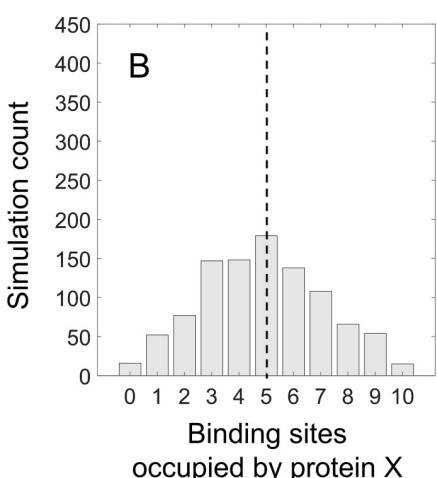 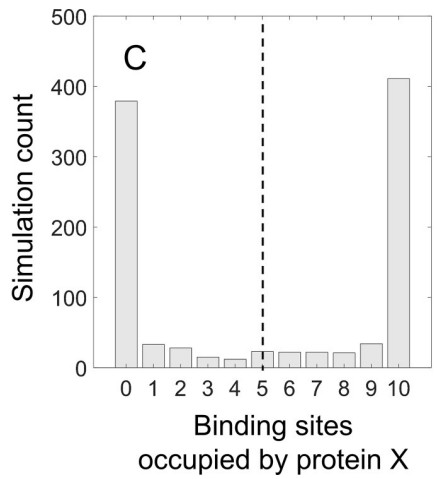

**Fig 6. Stochastic simulations and deterministic solutions for a cellulosome model system, in which two proteins X and Y expressed in a parallel gene circuit bind to 10 scaffold binding sites to form a complexoform.** The outcome of 1,000 stochastic simulations is indicated as grey bars, while the deterministic solution is indicated by a dashed line. The model parameters are summarized in Table 1, with the translation rate constant in this figure being varied across several orders of magnitude: 3.10$^{-5}$ s$^{-1}$ (A), 3.10$^{-2}$ s$^{-1}$ (B), and 3.10$^1$ s$^{-1}$ (C). For each simulation condition, the stochastic variation in cellulosome assembly is calculated as the standard deviation of the distribution of stochastic simulations. The discrepancy between the stochastic simulations and deterministic solution is calculated as the average difference between each simulation and the deterministic solution.

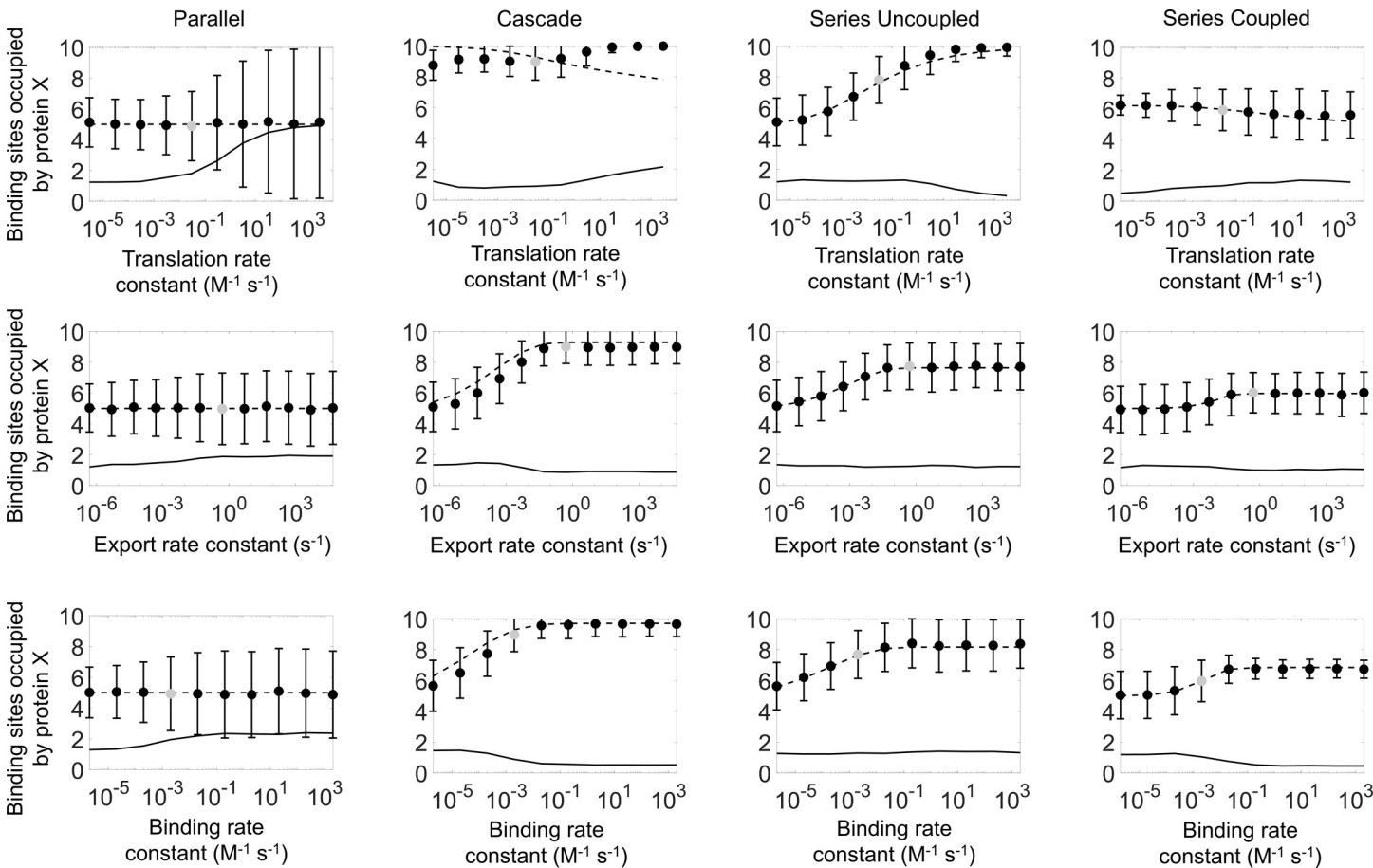

**Fig 7. Computational simulations of cellulosome assembly consisting of a scaffold with 10 available binding sites for proteins X and Y for four stochastically simulated genetic circuits.** The model parameters are summarized in Table 1. The vertical axis indicates the number of scaffold binding sites occupied by protein X. The average of 1,000 stochastic simulations is indicated by solid circles for each condition, with error bars indicating the standard deviation from the simulation. Deterministic solution results are indicated by dashed lines. The discrepancy between stochastic and deterministic solutions (calculated as the average difference between each simulation and the deterministic solution) is indicated by solid lines. Grey dots indicate the estimate of physiological rates (Table 1), about which each rate is varied.

with about 90% protein $X$; however, if the binding rate had been 10-fold slower (or the loss rate 10-fold higher), the fraction of protein $X$ in the assembly drops to 77%, and a 1000-fold change predicts a fraction of 56% $X$, i.e., almost equal amounts of $X$ and $Y$. Both protein binding and protein loss could depend on the exact extracellular conditions (e.g., the degree of crowding), thus impacting the assembly purely by altering the kinetic rates involved. Thus, our data demonstrate that both the kinetic rates and the genetic circuit can moderate the eventual assembly outcome in the case of the bacterial cellulosome.

## Discussion

In summary, our work shows that the underlying genetic circuit architecture does modulate the protein assembly. However, it is the interplay between the circuit architecture and the genetic and physical rate kinetics that together determine the protein assembly structure. We demonstrate two distinct behaviors of kinetic assembly: a slow equilibrium regime, in which the average assembly is well described by equilibrium statistical mechanics, and a fast non-equilibrium regime, in which the average assembly arrests before the system reaches equilibrium. Regardless of the equilibrium or non-equilibrium regime, the cumulative protein

concentrations (i.e., the total amount of protein available to bind over time) determine the eventual complexoform distribution (S5 Appendix). Furthermore, we demonstrate that the two regimes can be regulated by tuning any of the kinetic rates involved in the protein expression and assembly process, whether biochemical or physical. The arresting of the assembly into a non-equilibrium structure has previously been observed on much larger length scales, such as in the dynamic arrest occurring in macroscopic protein assemblies such as biopolymer networks, in which the kinetics of assembly can highly affect the non-equilibrium assembly structure [31–34]. Our work shows that similar dynamic arrest can occur on a much smaller scale as well, in the assembly of protein complexes involving just a handful of individual proteins. The results highlight new mechanisms, in addition to restrictive or preferential binding, through which systems can control stochastic processes such as protein assembly. The results also underscore the importance of both kinetics and stochastic non-equilibrium behavior in addition to the genetic circuit architecture as modulators of protein assembly processes.

There are several future considerations for our results. First, our work did not consider the details of how genetic context affects the transcription or translation rate of genes. For instance, a prior work [35] shows that the arrangement of two genes can affect the level, dynamic range, and sensitivity of their expression. The spatial arrangement of the two genes will affect the expression of our parallel genetic module. Specifically, if the parallel genetic module is inducible by certain chemicals, its sensitivity and dynamic range of induction would also be affected by the arrangement of the two genes, as suggested by the prior work. In our work, we have not considered the induction dynamics of the genetic module, which may serve as another regulatory force of complexoforms.

Second, one challenge in our work is to obtain the analytical solution of the stochastic equations. For instance, previous work by Laurenzi, Renyi, Mcquarrie, and Ishida [36–39] have solved the analytical solutions for A+B↔C. To obtain the solution, the authors exploit the conservation of mass between A, B, and C. Finding similar analytical solutions to our model involves two challenges. First, there are multiple bimolecular reactions in our model. Second, it is challenging to incorporate the conservation of mass for intermediate reactions of our model. There is, however, a qualitative agreement between our simulations and prior analytical solution. For instance, Laurenzi shows that stochastic solution deviates from a deterministic solution when the number of molecules is low. We observe this trend generally for all our results because the molecular number in our system is low. Furthermore, Laurenzi shows that the variance of molecular concentrations peaks at intermediate reaction time when the reaction becomes irreversible. Likewise, we observe that when the reaction rate becomes slower, our system takes a long time to approach the steady-state solution. The longer transient kinetics increase the duration when the system exhibits a high variance of molecular concentration. This enhanced variance likely leads to the significant deviation between the stochastic and deterministic solutions when reaction rate constants are high.

Third, the arrangement of two genes in the series coupled module may affect the expression level of the genes. Typically, the second gene is expressed at a lower level than the first gene. A prior work [40] has quantified the effect of GC content, size, and folding energy of the first gene on the expression of the second gene. Incorporating these details will improve the prediction power of our model on the distribution of complexoforms.

Our study points to the needs of quantifying heterogeneity of complexoform inside cells and studying the regulatory mechanisms of the heterogeneity. New mass spectrometry technology and modeling tools may be used to reveal new insights into complexoforms that were not possible before. For instance, the proteasomes of Rhodococcus consisted of both alpha ($\alpha_1$, $\alpha_2$) and beta ($\beta_1$, $\beta_2$) subunits [41]. The subunits can form four variants of complexoform ($\alpha_1\beta_1$, $\alpha_1\beta_2$, $\alpha_1\beta_2$, $\alpha_2\beta_1$) with similar proteasome functions. Does natural Rhodococcus control

the heterogeneity of the proteasome at the genetic or protein level, or both? The Trp synthase of *E. Coli* also consists of two α and two β subunits. The genes are arranged in the same order as the assembly order of the subunits [42]. Does the assembly process of the synthase fall in the equilibrium or non-equilibrium regimes? Answer to the question requires detailed measurements of the kinetic parameters that are not available yet. Finally, the Pili of *E. Coli* consists of four different protein subunits, and the gene order is arranged opposite of the assembly order [43]. Does this contradiction of gene and assembly order indicate that the assembly is solely determined by protein-protein binding and that there is less heterogeneity in the final protein complex? To this end, when the kinetic parameters of these processes become available, our modeling framework may be used to predict the heterogeneity of the complexoform and the equilibrium vs. non-equilibrium control of the systems. Furthermore, a meta-analysis of genome sequences and protein-protein interaction maps may be used to reveal the relative abundance of each circuit architectures for assembling known complexoforms.

## Methods

### Modeling protein expression, export, and binding

Our computational model consists of a set of coupled biochemical and physical processes, starting from transcription, and ending with the binding of protein products to 10 scaffold proteins, each with two docking sites to form a final complexoform. In the first process, an RNA polymerase binds to a promoter to synthesize mRNA. This transcription step is followed by the translation of two different protein products, denoted *X* and *Y*. The proteins are then transported for binding to a scaffold protein to form a three-protein complexoform. The model, which is summarized schematically in Fig 1, also incorporates the degradation of mRNA and diffusive loss of proteins after transport.

The transcription and translation of the two proteins occur through one of four genetic circuits: (1) a parallel expression circuit, in which transcription and translation of the two proteins happen concurrently and independently; (2) a cascade expression circuit, in which the transcription and translation of protein *X* create the needed transcription activator for the subsequent transcription of protein *Y*; (3) a series uncoupled expression circuit, in which the transcription of the two genes occurs sequentially but with uncoupled translation; and (4) a series coupled expression circuit, in which both the transcription and translation are sequential. Within each of these four different genetic circuit architectures, the rates of genetic expression and physical export and protein binding processes are varied in the model. In all cases, the evolution of the system is arrested as the docking sites are exhausted, and no further changes to the protein assembly structure can occur once they are all occupied.

The set of biochemical and physical processes defining each genetic circuit is listed in S1 Appendix. All processes are modeled as linear reactions, in which the reaction is linearly dependent on a rate constant and on the concentration of the reactants. Thus, for a set of reactions of the form

$$A + B + \cdots \xrightarrow{k_1} Q$$

$$Q + R + \cdots \xrightarrow{k_2} Z$$

the rate of change of any reactant is described by a first-order differential equation:

$$\frac{d[Q]}{dt} = +k_1 \cdot [A][B] \cdots - k_2 [Q][R] \cdots$$

Here, [·] indicates the concentration of each reactant. Any additional reactions involving the reactant $Q$ is incorporated as additional, linear terms, and the rate of change of each reactant is modeled by an additional differential equation. In this model, any arbitrary number of reactions involving any number of reactants can thus be written as a set of coupled, first-order differential equations. For large ensembles (i.e., large concentrations of all reactants involved), these coupled differential equations are deterministic, and the outcome for each reactant can be predicted exactly by solving the equations numerically. However, for a sufficiently small number of molecules, the outcome becomes stochastic and must be approached using stochastic algorithms. We, therefore, solve the coupled differential equations both deterministically to obtain exact numerical solutions, and stochastically using a stochastic algorithm.

## Computational model implementation

All computer simulations are done using Matlab (Mathworks, Natick MA). The built-in *ode45* differential equation solver is used for all deterministic numerical solutions. The initial conditions of the deterministic differential equation solutions are identical to those of the stochastic model, except that the amount of each reactant is treated as a continuous, rather than a discrete variable, and are allowed non-integer values.

Stochastic simulations use a custom-written Gillespie algorithm [44] in Matlab, in which each reaction outlined in S1 Appendix is treated stochastically. In this algorithm, each possible reaction is numerically simulated using a Monte Carlo technique by generating a tentative reaction time for each reaction based on the concentrations and reaction rates at that instant in time, and in each case, executing the reaction with the shortest tentative reaction time. This process is repeated until all binding sites are occupied, after which no further evolution of the protein assembly can occur. Unlike in the deterministic solution, the stochastic simulation only allows for discrete changes in the amount of each reactant, and only integer values are allowed for each reactant. To account for the stochastic variation between simulations, each set of conditions is simulated 1,000 times and averaged. For each set of initial conditions, we calculate the average assembly structure for the stochastic simulation by counting the docking site occupancy (i.e., the final average number of scaffold proteins with *XX*, *XY*, or *YY* count for each run of the simulation). The standard deviation is calculated for each of the three types of assemblies. We quantify the stochastic variation in the simulation as the sum of these three standard deviations.

To compare the deterministic and stochastic solutions, we calculate the difference between the amounts of each type of protein complex predicted from the deterministic solution with the average amount observed from the stochastic simulation. The sum of these three differences is used as a measure of the overall discrepancy between the deterministic and stochastic computational solutions.

For each genetic circuit, we vary the translation rate, the protein export rate, and the protein binding rate to investigate the impact of kinetics on the modulation of the final protein assembly. The initial conditions and rates are all listed in S2 Appendix. In order to investigate the impact of the relative, rather than the absolute, kinetic rates on the final assembly structure in the model system, simulations are done using arbitrary but dimensional units: time is indicated in units of t, concentrations are indicated in units of [C], and rates are indicated in units of $[C]^{-1}t^{-1}$ or $t^{-1}$.

## Biological rates of the cellulosome complex

In order to compare our simulation results in arbitrary units to biological systems, we run both deterministic and stochastic simulations using typical concentrations and biochemical

and physical rates of protein expression, export, and binding for the bacterial cellulose-degrading multi-enzyme complexes, known as cellulosomes. We modeled the assembly of cellulosomes, which comprise scaffold proteins containing repeated copies of Cohesin (Coh) domains and catalytic enzymes containing Dockerin (Doc) domains. Each Coh within the scaffold serves as a binding site for the corresponding Doc domain located at the C-terminus of each enzyme. The enzymes are secreted from the bacteria and assemble onto the extracellular scaffold proteins through Coh-Doc binding. The Coh-Doc interactions, albeit high-affinity, are promiscuous, and dozens of different enzymes containing homologous and closely related terminal Doc sequences are able to bind to any of the Coh domains within the scaffold. Although the exact cellulosome complex can contain many types of catalytic modules and scaffoldin binding sites [5], we adopt a model in which the external scaffold has 10 available sites to which two different proteins can directly bind, as this model captures the competing role of both timing and gene circuit architecture in modulating the complexoform. While some rates relevant to complexoform enzyme expression and assembly have been previously measured and reported in the literature, such as the promoter binding and unbinding rates [16] and the rates of the protein-protein binding interaction [25–26], we infer or estimate other rates to an order of magnitude, as described below.

Previous reports have shown that mRNA degradation occurs with half-lives ranging from a few minutes in *E. Coli* [21] and ~20 minutes in *S. Cerevisiae* yeast [45] to several hours in *H. Sapiens* cells [46]. Therefore, we assume a typical bacterial mRNA half-life of 5 minutes, corresponding to a decay rate of $0.002\ s^{-1}$.

The rate of protein export is assumed for on the size of 50–100 kDa [23], suggesting an effective diffusion coefficient of about $1\ \mu m^2 s^{-1}$ through a cytoplasmic environment [24]. We, therefore, estimate that diffusive transport in a bacterium with a size of ~2 μm would take on the order of 2 seconds, corresponding to a rate of export on the order of $0.5\ s^{-1}$ after completed expression.

We assume that RNA polymerase creates RNA at a rate of about 45 nucleotides per second, and that ribosomal translation occurs at a rate of about 15 amino acids per second [17–20,22]. Since 3 base pairs code for one amino acid, the rate of transcription and translation are roughly equal. Assuming a typical protein size of about 50 kDa and an average amino acid size of 110 Da, one full protein would take 30 seconds of transcription and translation, corresponding to a rate of $0.03\ s^{-1}$ for both processes.

In order to consider reaction rates and concentrations in terms of the number of molecules of each reactant inside a bacterium, we convert molar concentrations to molecules per bacteria volume with *E. Coli* as a reference. A concentration of 1 M corresponds to $6 \cdot 10^8$ molecules/bacterium, assuming a bacterial volume of 1 fL. We assume one copy of each gene (typically the case for bacteria with one chromosome), with about 100 RNA polymerases per gene [28] (estimated from ~23 RNA polymerase per lacZ gene). Although there are about 20,000 ribosomes in one *E. Coli* bacterium [29], these are employed in the expression of roughly 4,000 different genes [30]; we, therefore, assume 5 available ribosomes.

It is difficult to estimate the exact external per-molecule binding rate and loss rate of individual proteins. The cohesin-dockerin association rate has been measured to be about $1 \cdot 10^6$ $M^{-1}s^{-1}$ [25–26], and assuming that the outside reaction volume is similar to the volume of a single bacterium, the external binding rate is $0.002\ s^{-1}$ per molecule. Protein lifetimes are typically on the order of tens of minutes or several hours [27]. Assuming a lifetime of about an hour, this implies a protein loss rate of $0.0002\ s^{-1}$ per molecule.

All the values used for the cellulosome assembly simulations are summarized in Table 1. Since most values are estimated to an order of magnitude, we vary the translation, export, and binding rates across many orders of magnitude to elucidate whether the cellulosome

complexoform assembly process occurs in the equilibrium or non-equilibrium regime. In the simulation, we vary the ratio of the protein association rate and diffusive loss rate. Although data shows a slow dissociation of the dockerin/cohesin complex, we assume that the binding is irreversible to allow us to study the structure of the completed assembly right after protein expression, transport, and assembly. We note that with a slowly reversible binding, the complexoform distribution should eventually equilibrate with the external solution concentrations of protein in solution over long time scales.

All data generated for and presented in this study are available in full in the S1 Data.

## Supporting information

**S1 Appendix.**
(PDF)

**S2 Appendix.**
(PDF)

**S3 Appendix.**
(PDF)

**S4 Appendix.**
(PDF)

**S5 Appendix.**
(PDF)

**S1 Data.**
(XLSX)

## Author Contributions

**Conceptualization:** Michael A. Nash, Cheemeng Tan.

**Formal analysis:** Mikkel Herholdt Jensen, Eliza J. Morris, Cheemeng Tan.

**Funding acquisition:** Michael A. Nash, Cheemeng Tan.

**Investigation:** Mikkel Herholdt Jensen, Eliza J. Morris, Hai Tran.

**Methodology:** Mikkel Herholdt Jensen, Eliza J. Morris, Cheemeng Tan.

**Software:** Mikkel Herholdt Jensen, Eliza J. Morris, Hai Tran, Cheemeng Tan.

**Validation:** Mikkel Herholdt Jensen, Eliza J. Morris, Cheemeng Tan.

**Writing – original draft:** Mikkel Herholdt Jensen, Cheemeng Tan.

**Writing – review & editing:** Mikkel Herholdt Jensen, Eliza J. Morris, Hai Tran, Michael A. Nash, Cheemeng Tan.

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
