## [Decision Letter · Decision Letter 0]

21 Feb 2020

Dear Dr. Tan,

Thank you very much for submitting your manuscript "Stochastic ordering of complexoform protein assembly by genetic circuits" for consideration at PLOS Computational Biology. As with all papers reviewed by the journal, your manuscript was reviewed by members of the editorial board and by several independent reviewers. The reviewers appreciated the attention to an important topic. Based on the reviews, we are likely to accept this manuscript for publication, providing that you modify the manuscript according to the review recommendations.

In addition to the reviewer comments below, please also consider my own comment that this work seems to be related to earlier work linked to here --  https://journals.plos.org/plosone/article?id=10.1371/journal.pone.0032032 -- which also used a simulation-based approach to determine distribution of complexoforms and their kinetics. If so, it may be worth citing and discussing.

Sincerely,

James R. Faeder

Associate Editor

PLOS Computational Biology

William Noble

Deputy Editor

PLOS Computational Biology

[LINK]

Reviewer's Responses to Questions

**Comments to the Authors:**

Reviewer #1: This is a nice study elucidating the role of biological circuitry in control and assembly of complexoform proteins. With a few modifications, it should be accepted for publication. Please take the following suggestions into account:

• The authors should take into account the effect of different promoter strengths and different ribosome binding site strengths for each separate gene. The authors should analyze the effect of changing the promoter and RBS strengths for an individual gene, and compare this to the effect of changing the genetic circuit architecture. If circuit architecture effects tend to dominate over promoter and RBS binding site effects, this should be explained.

• The authors should include some discussion of the relative abundance of each type of circuit architecture found in natural genomes, for assembling known complexoforms. Is there a particular circuit layout which tends to appear more often than the others?

• The authors should discuss their work in relation to the following literature:

o Yeung et al. Biophysical Constraints Arising from Compositional Context in Synthetic Gene Networks. Cell Systems. 2017.

o In particular, the authors should include a discussion of how the orientation of genetic circuits in a parallel configuration affects transcription, and how this might impact their model.

• The term “complexoform” is of limited use in the literature (a Google Scholar search returns four relevant results). Accordingly, in order to assure clarity, the authors should introduce the term as “complex isoforms” as is done in Fonslow et al.

Reviewer #2: This is an interesting and novel study, where computational simulations are used to analyze a recently appreciated phenomenon -- complexoforms. In particular, the authors thoroughly analyzed how gene circuit architecture and rate constants can affect distribution of complexoforms. They have also thoroughly compared deterministic and stochastic simulations to examine the potential contribution of noise in this dynamics.

The study is well designed and the paper is well written and easy to follow. The model formulation and subsequent analysis are fairly standard. But the key innovation of the study is the application context (to a novel problem, as far as I can tell). The work defines a useful blueprint for the dynamic analysis of complexoforms. The modeling framework (in particular) and the results can be useful for guiding future experimental analysis.

I am in support of the publication of the work with some minor clarifications and revisions, as detailed below.

Specific points:

1. In essence, the different circuit architectures dictate the temporal dynamics of X and Y, which in turn bind the the different scaffold sites. As such, it is not entirely surprising that different architectures would modulate the find distribution of complexoforms differently. I wonder if it is possible to deduce some more general rules that govern the final distribution based on the temporal profiles of X and Y (outside the cell).

2. Related to the point above, the binding kinetics of X and Y to scaffold site should be important as well. If I understand it correctly, the authors assume the binding to be irreversible (Table 1). I wonder what happens to the distributions if the binding is reversible, which appears to be more plausible.

3. Does the number of binding sites matter? At it stands, the choice of the number seems ad hoc.

4. There was a lot of focus on comparing deterministic modeling and stochastic modeling. The resulting insights are not well articulated and need to be clarified in a revision. Is there experimental data that the modeling results can be compared to?

Reviewer #3: In the manuscript titled " Stochastic ordering of complexoform protein

assembly by genetic circuits", Jensen et al. used combination of stochastic

simulation and deterministic differential equation model to investigate how

parameters such as timing of transcription, translation, and transport rate

impact formation preferences of complexoform. They found that in two different

extreme scenarios, complexoform variations are also vastly different. At the

end, the authors used a real biological example to verify their framework. I

think the authors did a good job explaining this relatively new topic and made

the text fairly easy to follow. Discussing a relatively less studied new area,

this work used standard simulation methods to shed some new light on possible

mechanisms controlling complexoform preferences, attributing some of the

control to gene circuit layout. This finding hopefully would inspire more

future work in this area to investigate this interesting and important problem.

I found the manuscript to be informative, the

approaches used appropriate, and would recommend for its publication given my

following concerns/comments to be addressed:

1. One major concern of mine is the lack of any analytical results. Numerical

simulations are definitely preferred in many scenarios, including this work.

But careful analysis often is also needed. It would complement numerical

simulations well and often offer a fuller picture. I understand analytical

results are not possible for complex systems, but in this case, there is no

feedback and an analysis of the simplified scenarios should be within reach.

2. I am curious why the authors did not consider mRNA degradation as one key

parameters. mRNA degradation has been shown to be a key regulatory mechanism

in biology. In addition, recent work (Wu et al, Cell Systems 6, 206-215, 2018)

showed that in the series coupled scenarios, the order of the gene X and Y

would affect their stability and hence could cause a change in the

complexoform formation biases.

**Have all data underlying the figures and results presented in the manuscript been provided?**

Reviewer #1: Yes

Reviewer #2: Yes

Reviewer #3: Yes

PLOS authors have the option to publish the peer review history of their article (what does this mean?). If published, this will include your full peer review and any attached files.

Reviewer #1: No

Reviewer #2: No

Reviewer #3: No
---

## [Decision Letter · Decision Letter 1]

28 May 2020

Dear Dr. Tan,

We are pleased to inform you that your manuscript 'Stochastic ordering of complexoform protein assembly by genetic circuits' has been provisionally accepted for publication in PLOS Computational Biology.

Best regards,

James R. Faeder

Associate Editor

PLOS Computational Biology

William Noble

Deputy Editor

PLOS Computational Biology

Reviewer's Responses to Questions

**Comments to the Authors:**

Reviewer #1: The authors have adequately addressed my points.

Reviewer #2: The authors have fully addressed my raised points and I support the publication of the paper.

**Have all data underlying the figures and results presented in the manuscript been provided?**

Reviewer #1: Yes

Reviewer #2: Yes

PLOS authors have the option to publish the peer review history of their article (what does this mean?). If published, this will include your full peer review and any attached files.

Reviewer #1: No

Reviewer #2: No

---

## [Editor Report · Acceptance letter]

19 Jun 2020

PCOMPBIOL-D-20-00101R1 

Stochastic ordering of complexoform protein assembly by genetic circuits

Dear Dr Tan,

I am pleased to inform you that your manuscript has been formally accepted for publication in PLOS Computational Biology. Your manuscript is now with our production department and you will be notified of the publication date in due course.

With kind regards,

Sarah Hammond
